

# Comparison of pediatric scoring systems for mortality in septic patients and the impact of missing information on their predictive power: a retrospective analysis

Christian Niederwanger[1], Thomas Varga[2], Tobias Hell[3], Daniel Stuerzel[4], Jennifer Prem[4], Magdalena Gassner[4], Franziska Rickmann[4], Christina Schoner[4], Daniela Hainz[4], Gerard Cortina[1], Benjamin Hetzer[1], Benedikt Treml[4] and Mirjam Bachler[4,5]

[1] Department of Pediatrics, Pediatrics I, Medical University of Innsbruck, Innsbruck, Austria
[2] Institute of Anaesthesiology, University of Zurich and University Hospital Zurich, Zurich, Switzerland
[3] Department of Mathematics, Faculty of Mathematics, Computer Science and Physics, University of Innsbruck, Innsbruck, Austria
[4] Department of General and Surgical Critical Care Medicine, Medical University of Innsbruck, Innsbruck, Austria
[5] Department of Sports Medicine, Alpine Medicine and Health Tourism, UMIT - University for Health Sciences, Medical Informatics and Technology, Hall in Tyrol, Austria

Corresponding author
Mirjam Bachler,
mirjam.bachler@tirol-kliniken.at

## ABSTRACT

**Background**. Scores can assess the severity and course of disease and predict outcome in an objective manner. This information is needed for proper risk assessment and stratification. Furthermore, scoring systems support optimal patient care, resource management and are gaining in importance in terms of artificial intelligence.

**Objective**. This study evaluated and compared the prognostic ability of various common pediatric scoring systems (PRISM, PRISM III, PRISM IV, PIM, PIM2, PIM3, PELOD, PELOD 2) in order to determine which is the most applicable score for pediatric sepsis patients in terms of timing of disease survey and insensitivity to missing data.

**Methods**. We retrospectively examined data from 398 patients under 18 years of age, who were diagnosed with sepsis. Scores were assessed at ICU admission and re-evaluated on the day of peak C-reactive protein. The scores were compared for their ability to predict mortality in this specific patient population and for their impairment due to missing data.

**Results**. PIM (AUC 0.76 (0.68–0.76)), PIM2 (AUC 0.78 (0.72–0.78)) and PIM3 (AUC 0.76 (0.68–0.76)) scores together with PRSIM III (AUC 0.75 (0.68–0.75)) and PELOD 2 (AUC 0.75 (0.66–0.75)) are the most suitable scores for determining patient prognosis at ICU admission. Once sepsis is pronounced, PELOD 2 (AUC 0.84 (0.77–0.91)) and PRISM IV (AUC 0.8 (0.72–0.88)) become significantly better in their performance and count among the best prognostic scores for use at this time together with PRISM III (AUC 0.81 (0.73–0.89)). PELOD 2 is good for monitoring and, like the PIM scores, is also largely insensitive to missing values.

**Conclusion**. Overall, PIM scores show comparatively good performance, are stable as far as timing of the disease survey is concerned, and they are also relatively stable in terms of missing parameters. PELOD 2 is best suitable for monitoring clinical course.

## INTRODUCTION

Early detection of critically ill patients is essential for timely, good care in a suitable facility. Sepsis remains one of the leading causes of childhood death, although our understanding of the pathophysiology of sepsis has changed drastically in the last couple of decades due to the development of new diagnostic projections and strategies in the treatment of this complex illness (*Dellinger et al., 2013*).

To help assess severity of illness for risk stratification in terms of required resources, stratify patients prior to randomization in clinical trials, compare intra- and inter-institutional outcome and survival, improve quality assessment as well as cost-benefit analysis, and to facilitate clinical decision making, prognostic scoring systems were established in the 1980s and have been improved and validated since (*Lemeshow & Le, 1994*; *Marcin et al., 1998*).

The first scoring systems were developed for adults and were less suitable for use in children. Finally, corresponding scores were presented specially for children and continuously developed and improved (*Leteurtre et al., 1999*; *Pollack, Ruttimann & Getson, 1988*; *Shann et al., 1997*). Some of them permit the probability of survival to be estimated as a function of the determined score. Today's scores, which are especially suitable for children, are, for example, the Pediatric Risk of Mortality (PRISM) score, from which its further developments, namely the PRISM III and PRISM IV scores, the Pediatric Index of Mortality (PIM) score, were derived, the PIM2 and PIM3 scores and the PELOD (Pediatric Logistic Organ Dysfunction) score followed by the PELOD 2 score (*Leteurtre et al., 2006*; *Leteurtre et al., 2013*; *Leteurtre et al., 1999*; *Pollack et al., 2016*; *Pollack, Patel & Ruttimann, 1996b*; *Pollack, Ruttimann & Getson, 1988*; *Shann et al., 1997*; *Slater, Shann & Pearson, 2003*; *Straney et al., 2013*). Only few studies deal with the question whether the individual scores are equally suitable for all types of conditions (*Dewi et al., 2019*; *Gemke & van Vught, 2002*; *Leteurtre et al., 2001*; *Muisyo et al., 2019*; *Qiu et al., 2017b*).

For each score, the development identified specific times or timescales for patient enrollment, within which the score provides the most accurate indication of the patient's condition and the likelihood of survival (*Leteurtre et al., 2006*; *Leteurtre et al., 2013*; *Leteurtre et al., 1999*; *Pollack et al., 2016*; *Pollack, Patel & Ruttimann, 1996b*; *Pollack, Ruttimann & Getson, 1988*; *Shann et al., 1997*; *Slater, Shann & Pearson, 2003*; *Straney et al., 2013*). However, some patients develop certain life-threatening conditions—such as sepsis—only during their inpatient stay and are actually hospitalized for a completely different reason, for example following a surgical intervention (*Sidhu et al., 2015*). In such a case, it is obvious that the condition of the patient determined at admission can only conditionally predict the course of a complication developed at a later time. Although some scores consider the admission reason in their evaluation (*Pollack et al., 2016*; *Straney et al., 2013*), the further course is still open.

Scores are calculated based on vital signs, laboratory parameters and other patient parameters. In everyday clinical practice and of course as a consequence of the retrospective study design it is often not possible to determine all required data, because they are not recorded, not collected or can no longer be found. However, incomplete data raise the question of score accuracy. Some evidence suggests that knowing the completeness of necessary data is essential for correct score results (*Agor et al., 2019*; *Gorges et al., 2018*).

This knowledge of data completeness and its impact on the results is becoming of even more interest with regard to the keyword "artificial intelligence." More and more artificial intelligence assessments are based on patient stratification and different kinds of scores (*Abbasi, 2018*; *Komorowski et al., 2018*). For this reason, it is wise to also know more about the influence of "missing values" on the accuracy of the informative value of the scores, since not all parameters needed for creation of the scores are available or measured.

In this study, the usual pediatric scores are compared in terms of their predictive value in a group of septic children. Also, the optimal timing for determining these scores in the clinical course picture of sepsis was evaluated in addition to the influence of the lack of data on the predictive value of the different scores.

## METHODS

This retrospective analysis included 398 critically ill pediatric patients treated at Innsbruck Medical University Hospital.

### Inclusion of patients

The medical files of patients younger than 18 years of age with diagnosed sepsis or a proven blood stream infection between 2000 and 2019 were reviewed. Children fulfilling the definitions according to the International Pediatric Consensus Conference (*Goldstein, Giroir & Randolph, 2005*) were included. The current definition of pediatric sepsis is systemic inflammatory response syndrome (SIRS) in the presence of or as a result of suspected or proven infection (*Goldstein, Giroir & Randolph, 2005*). SIRS is given when at least two of the four criteria are present, one of which must be abnormal temperature or leukocyte count (*Goldstein, Giroir & Randolph, 2005*). In this connection, the fulfillment of SIRS criteria is dependent on the age-specific normal values. The study protocol was approved by the institutional review board of the Medical University of Innsbruck (AN2013-0044 and EK Nr: 1109/2019).

### Data collection

We collected the demographic variables such as age, sex and the diagnosed underlying disease. The underlying disease was assigned to the appropriate organ category: central nervous system, cardiovascular system, respiratory system, hepatic or renal failure. Also recorded was whether the patient suffered from an oncologic disease. Furthermore, we collected routinely measured laboratory parameters on the day of peak C-reactive protein.

C-reactive protein was chosen as parameter for the most severe stage of sepsis since it reflects the inflammatory process and is widely used in clinical routine. Many studies have described an interrelation between an elevated C-reactive protein level and sepsis

(*Koozi, Lengquist & Frigyesi, 2020*; *Maury, 1989*; *Povoa et al., 1998*; *Presterl et al., 1997*; *Schentag et al., 1984*) and that C-reactive protein is highest at the most severe point of sepsis (*Castelli et al., 2004*; *Lobo et al., 2003*; *Povoa et al., 1998*; *Povoa et al., 2005*). Furthermore, elevated C-reactive protein is also associated with organ failure (*De Beaux et al., 1996*; *Ikei, Ogawa & Yamaguchi, 1998*; *Pinilla et al., 1998*; *Rau et al., 2000*; *Waydhas et al., 1996*), which makes C-reactive protein a suitable parameter for the surveillance of sepsis severity.

PRISM, PRISM III, PRISMIV, PIM, PIM2, and PIM3 as well as PELOD and PELOD2 scores were retrospectively assessed. Since in the realm of this study the scores were evaluated for their ability to depict the disease process and etiology, we have chosen two time points for score assessment, namely the day of admission as well as the day of peak C-reactive protein (File S1). In this way we were able to analyze whether the time of assessment influences the predictive power of the individual scores. In-hospital mortality and multi-organ dysfunction syndrome (MODS) were chosen as outcome parameters. During data acquisition, the percentage of missing values was recorded for each score as a function of the number of their requested parameters.

## Statistical analysis

A mathematician (TH) not involved in the study procedures or patient assessment was responsible for the statistical analyses conducted using R, version 3.5.3. We present continuous data as median (25th to 75th percentile) and categorical variables as frequencies (%). We show effect size and precision with estimated median differences between survivors and non-survivors for continuous data and odds ratios (OR) for binary variables, with 95% CIs. All statistical assessments were two-sided, and a significance level of 5% was used. We applied the Wilcoxon rank sum test and Fisher's exact test to assess differences between survivors and non-survivors.

The precision of the scores as the difference between predicted mortality and actual mortality is presented depending on the percentage of missing parameters. With respect to their diagnostic ability, the scores were compared by means of ROC curves, and DeLong's test was used to assess differences in ROC AUC. Corresponding 95% CIs were provided for the ROC AUC of the scores and the differences in the ROC AUC between scores. For this analysis only complete data were used.

## RESULTS

### Patient characteristics

For this analysis 398 patients met the eligibility criteria for study inclusion. In-hospital mortality in these septic children was 13.6% ($n = 54$). The median age of the children was 29.6 months, whereas 14.6% of the study population consisted of neonates. There was no difference in survival between males and females (see Table 1).

The most affected organ in terms of underlying disease was the respiratory system in 26.8% of the children followed by diseases of the central nervous system (22.3%) and the cardiovascular system (21.6%). Only the rate of kidney failure was significantly higher in the non-survivors, whereas the proportion of affected central nervous systems and digestive tracts tended to be higher in the non-survivors as well.

**Table 1  Characteristics[a] of patients stratified for survival and non-survival.**

| | Total ($n = 398$) | Non-survivors ($n = 54$) | Survivors ($n = 344$) | Estimate with 95% CI[b] | p value[c] |
|---|---|---|---|---|---|
| Female gender | 176/398 (44.2%) | 22/54 (40.7%) | 154/344 (44.8%) | 1.18 (0.63 to 2.22) | 0.6591 |
| Age[d] (months) | 29.6 (3.83–105.64) | 22.41 (1.22–88.99) | 30.68 (4.04–107.54) | −1.1 (−10.81 to 5.3) | 0.4817 |
| Neonates <1 month | 58/396 (14.6%) | 12/54 (22.2%) | 46/342 (13.5%) | 0.54 (0.26 to 1.22) | 0.0988 |
| Infants 1–3 months | 35/396 (8.8%) | 4/54 (7.4%) | 31/342 (9.1%) | 1.25 (0.41 to 5.06) | 1 |
| **Predicted mortality (%) at ICU admission** | | | | | |
| PRISM | 10.53 (1.54–7.55) | 19.79 (2.3–20.75) | 9.17 (1.5–7) | 10.62 (1.3 to 19.94) | 0.0264 |
| PRISM III | 6.12 (0.05–1.73) | 17.76 (0.39–8.62) | 4.36 (0.05–1.04) | 13.4 (2.92 to 23.87) | 0.0135 |
| PRISM IV | 3.59 (0.05–0.71) | 12.09 (0.12–2.71) | 2.31 (0.05–0.53) | 9.79 (0.82 to 18.76) | 0.0333 |
| PIM | 6.28 (1–5) | 16.06 (2.84–23) | 4.83 (1–4) | 11.23 (4.42 to 18.04) | 0.0017 |
| PIM 2 | 6.23 (0.75–4.01) | 16.64 (2.8–17.1) | 4.69 (0.75–3.7) | 11.95 (4.46 to 19.44) | 0.0024 |
| PIM 3 | 7.72 (1.11–4.16) | 18.93 (2.88–15.3) | 6.06 (1.11–3.53) | 12.86 (4.28 to 21.44) | 0.0041 |
| PELOD | 16.24 (0.96–16.25) | 32.23 (1.2–79.58) | 13.9 (0.96–16.25) | 18.33 (6.17 to 30.5) | 0.004 |
| PELOD 2 | 5.39 (0.13–2.21) | 18.21 (0.87–17.59) | 3.48 (0.13–1.39) | 14.72 (5.88 to 23.56) | 0.0016 |
| **Diagnosed underlying disease** | | | | | |
| Central nervous system | 81/364 (22.3%) | 15/44 (34.1%) | 66/320 (20.6%) | 0.5 (0.24 to 1.07) | 0.0532 |
| Cardiovascular | 78/361 (21.6%) | 14/42 (33.3%) | 64/319 (20.1%) | 0.5 (0.24 to 1.1) | 0.0704 |
| Digestive tract | 67/364 (18.4%) | 13/44 (29.5%) | 54/320 (16.9%) | 0.49 (0.23 to 1.08) | 0.0595 |
| Respiratory system | 97/362 (26.8%) | 17/43 (39.5%) | 80/319 (25.1%) | 0.51 (0.25 to 1.06) | 0.0649 |
| Oncologic | 51/365 (14%) | 9/43 (20.9%) | 42/322 (13%) | 0.57 (0.24 to 1.44) | 0.1636 |
| Kidney | 45/362 (12.4%) | 10/44 (22.7%) | 35/318 (11%) | 0.42 (0.18 to 1.04) | 0.0467 |
| Liver | 31/363 (8.5%) | 3/44 (6.8%) | 28/319 (8.8%) | 1.31 (0.38 to 7.05) | 1 |
| Skin | 18/361 (5%) | 1/43 (2.3%) | 17/318 (5.3%) | 2.37 (0.35 to 101.43) | 0.7078 |
| Other diagnoses | 108/367 (29.4%) | 16/44 (36.4%) | 92/323 (28.5%) | 0.7 (0.35 to 1.45) | 0.2931 |

Notes.

[a]Binary data are presented as no./total no. (%), continuous data as medians (25th–75th percentile), means for predicted mortality.

[b]Odds ratio for binary variables and estimated median difference for continuous variables, mean difference for predicted mortality.

[c]Differences between survivors and non-survivors assessed with Fisher's exact test for binary variables and the Wilcoxon rank sum test for continuous variables, the Welch two sample $T$ test for predicted mortality.

[d]For one survivor the exact age in months is not known.

## Evaluation of scores for the predictive ability for mortality

As seen in Fig. 1, the best prediction abilities in our study are seen for PIM (0.76; 0.68 to 0.76), PIM2 (AUC 0.78; 0.72 to 0.78) and PIM3 (AUC 0.76; 0.68 to 0.76), although there is no significant difference between them and the other tested scores except for PRISM. PRISM shows the poorest mortality prediction of all tested scores and is significantly poorer than PRISMIII ($p = 0.0122$), PIM ($p = 0.0059$), PIM2 ($p = 0.0125$) and PELOD2 ($p = 0.0359$). Also, the predictive ability of the scores PRISMIV and PELOD is as poor as that of PRISM, although with a slightly higher AUC. The most recent PRISMIV and PIM3 scores show no improvement in mortality prediction. On the contrary, they even show a deterioration in predictive value in our specific septic population as compared to the predecessor scores.

No difference was seen in thromboembolic complications or bleeding events between survivors and non-survivors (Table 2). The parameters for organ function with regard to renal or hepatic impairment also show no difference. Furthermore, in this septic

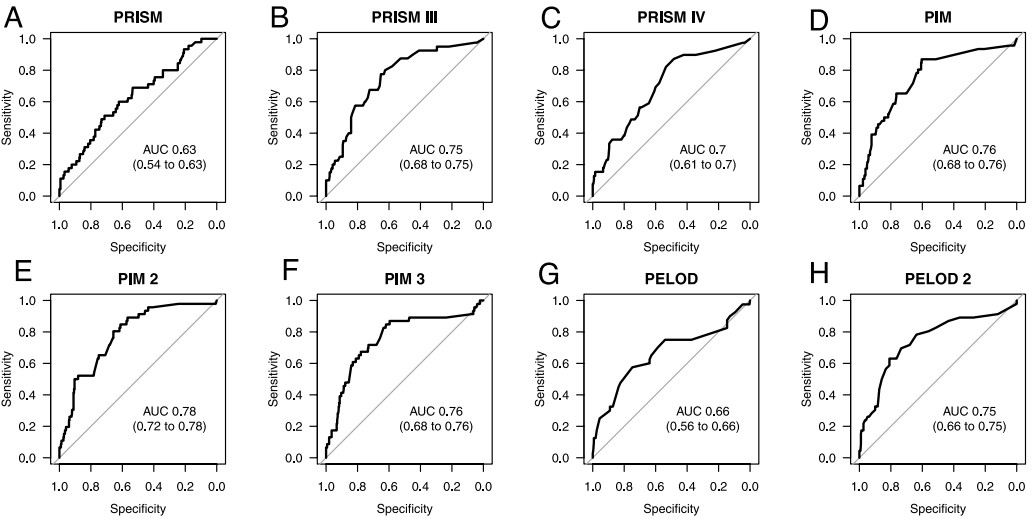

**Figure 1** **AUC ROC analysis of admission scores.** Receiver operating characteristic (ROC) curves and area under the curve (AUC) for mortality prediction at admission based on the Pediatric Risk of Mortality scores PRISM (A), PRISM III (B), and PRISM IV (C), the Pediatric Index of Mortality scores PIM (D), PIM 2 (E), and PIM 3 (F), as well as the Pediatric Logistic Organ Dysfunction scores PELOD (G), and PELOD 2 (H). The numbers in the brackets indicate the 95% CIs for the AUC.

population none of the recorded inflammatory parameters, namely C-reactive protein, procalcitonin, and interleukin-6, differentiate between survivors and non-survivors. Only the coagulation parameters show different values depending on the survival of the septic children. Fibrinogen, antithrombin and platelets were significantly higher in the survivors. As seen in the global coagulation tests, prothrombin time (PT; Quick) and activated thromboplastin time (aPTT), the patients who did not survive were in a hypocoagulable state. This is also reflected in the statistically larger number of bleeding complications seen in the non-survivors.

## Admission versus peak C-reactive protein: does the time of score evaluation matter?

To address the next question, namely whether the time of scoring makes a difference, two time points were compared: admission and the time when sepsis was most severe according to peak C-reactive protein. Except for PELOD, all scores improved towards the peak in C-reactive protein, as seen from their AUCs in Table 3. PRISMIV and PELOD2 even improved significantly and became, together with PRISMIII, the scores with the highest predictive ability, as seen in their AUC of 0.8, 0.84 and 0.81, respectively. The worst performance at this time was seen for PRISM followed by PELOD and PIM3.

## Missing values

Due to the nature of a retrospective design and the non-availability of all data for scoring, we checked whether there is an influence on the predictive ability of the different scores. For this purpose, we compared the actually observed mortality and the individual mortality

**Table 2  Outcome parameters[a] at peak C-reactive protein stratified for survival and non-survival.**

| | Total (n = 398) | Non-survivors (n = 54) | Survivors (n = 344) | Estimate with 95% CI[b] | p value[c] | Not known[d] |
|---|---|---|---|---|---|---|
| Thromboembolic event | 14/398 (3.5%) | 2/54 (3.7%) | 12/344 (3.5%) | 0.94 (0.2 to 8.89) | 1 | 0/0 |
| Bleeding complication | 20/398 (5%) | 9/54 (16.7%) | 11/344 (3.2%) | 0.17 (0.06 to 0.48) | 0.0004 | 0/0 |
| Creatinine [mg/dl] | 0.45 (0.3–0.72) | 0.58 (0.37–0.93) | 0.43 (0.3–0.71) | 0.12 (−0.01 to 0.25) | 0.0645 | 21/92 |
| Bilirubin [mg/dl] | 1.05 (0.44–4.49) | 1.78 (0.74–8.01) | 1 (0.43–4.16) | 0.56 (−0.07 to 2) | 0.0761 | 28/201 |
| D-dimer [μg/l] | 1421 (547.5-4595) | 1506 (310–8037) | 1402.5 (574-4314.5) | −96.59 (−589.88 to 728) | 0.6278 | 17/170 |
| Fibrinogen [mg/dl] | 462 (314–632) | 283.5 (186–476.25) | 484 (337–650) | −165 (−237 to −88) | 0.0001 | 14/119 |
| Antithrombin [%] | 63 (46.5–80.5) | 50.5 (41.25–64.75) | 65 (48–82) | −13 (−21 to −4) | 0.0033 | 16/147 |
| Activated partial thromboplastin time (aPTT) [s] | 46 (37–57.5) | 63.5 (50.75–76.25) | 45 (36–54) | 18 (11 to 25) | <0.0001 | 20/123 |
| Prothrombin time (PT, Quick) [%] | 68 (50–84) | 49 (36–71.5) | 71.5 (55–85) | −19 (−27 to −11) | <0.0001 | 19/122 |
| C-reactive protein [mg/dl] | 15.23 (6.44–26.18) | 16.83 (7.04–33.21) | 14.85 (6.44–25.15) | 2.74 (−0.86 to 7.6) | 0.1431 | 2/0 |
| Procalcitonin [mg/dl] | 11.3 (2.41–40.31) | 9.15 (4.45–39.48) | 11.71 (1.9–39.95) | 1.26 (−6.65 to 7.63) | 0.6081 | 36/221 |
| Hemoglobin [g/l] | 110.5 (97–130.75) | 113 (89–126.5) | 110 (97–131) | −4 (−22 to 14) | 0.7122 | 43/221 |
| Platelets [G/l] | 145 (60–250.5) | 53 (25–103) | 163 (81–260) | −86 (−119 to −54) | <0.0001 | 1/10 |
| Neutrophils [G/l] | 7.45 (3.62–14.05) | 9.9 (5.27–29.42) | 7.1 (3.6–13.33) | 3.5 (−0.3 to 8.4) | 0.0853 | 34/182 |
| Interleukin 6 [ng/l] | 120.2 (34–344.35) | 584.85 (342.27–827.42) | 120.2 (33.5–276.4) | 117.22 (−358.2 to 1049.3) | 0.3309 | 52/319 |
| Leukocytes [G/l] | 11.1 (5.9–17.85) | 8.45 (1.87–22.43) | 11.4 (6.15–17.45) | −1.1 (−4.4 to 1.8) | 0.4045 | 6/9 |
| Base excess arterial | −1 (−4.05–1.25) | −1 (−6.22–2.7) | −1 (−3.9–0.9) | −0.2 (−3 to 1.8) | 0.7468 | 28/223 |

**Notes.**
[a] Data are presented as medians (25th–75th percentile).
[b] Estimated median difference.
[c] Differences between survivors and non-survivors assessed with the Wilcoxon rank sum test.
[d] Number of missing measurements in survivors/non-survivors.

predictions as well as the AUCs depending on the different accepted extent of missing values.

Comparison of predicted versus actual mortality starts with only patients having no missing values for scoring, as seen in Fig. 2. The more missing values are accepted for scoring, the more patients are included, until 100% of the total patient population is included for scoring independently of the extent of their missing values.

As expected, depending on the size of the analyzed population, the line depicting the difference in mortalities settles only at a certain population size. When only those patients are included who have few missing values the patient number is very small, too small to make a validated statement about the difference in predicted and observed mortality.

With the increasing number of missing values allowed, all scores underestimate the actual mortality, except the PELOD score. The fewer missing parameters are accepted, the more similar the predicted and the actual mortality. Exceptions here are the PELOD and the PRISM scores as well as the PIM3 score with a high negative influenceability due to missing parameters, whereby the small number of cases limits the statement. Also, when comparing AUCs the small sample size is limiting, especially in PIM3.

Niederwanger et al. (2020), *PeerJ*, DOI 10.7717/peerj.9993

Peer∫

**Table 3  ROC analysis of scores predicting mortality.[a]** Results of the score analyses for mortality prediction at hospital admission and on the day with the highest level of C-reactive protein (CRP). The dark grey fields show the AUC with 95% CI. The numbers in the fields above the dark grey fields give the estimated difference in ROC curves with 95% CI and the fields below show the correlation with the corresponding ROC curves. Red *p* values indicate a significant difference. Only completed cases with all scores available are included.

*Left block p = 0.211; right block p = 0.2778*

| | PRISM | PRISMIII | PRISMIV | PIM | PIM2 | PIM3 | PELOD | PELOD2 | PRISM CRP | PRISMIII CRP | PRISMIV CRP | PIM CRP | PIM2 CRP | PIM3 CRP | PELOD CRP | PELOD2 CRP |
|---|---|---|---|---|---|---|---|---|---|---|---|---|---|---|---|---|
| **PRISM** | 0.6 (0.49 to 0.72) | −0.13 (−0.23 to −0.03); p = 0.0122 | −0.08 (−0.18 to 0.01); p = 0.0857 | −0.16 (−0.27 to −0.04); p = 0.0059 | −0.15 (−0.27 to −0.03); p = 0.0125 | −0.1 (−0.23 to 0.02); p = 0.1023 | −0.09 (−0.19 to 0.02); p = 0.1001 | −0.12 (−0.24 to −0.01); p = 0.0359 | **−0.06 (−0.13 to 0.01);** p = 0.1166 | −0.21 (−0.3 to −0.11); p = 0 | −0.19 (−0.28 to −0.11); p = 0 | −0.17 (−0.27 to −0.06); p = 0.0014 | −0.16 (−0.27 to −0.06); p = 0.0026 | −0.13 (−0.24 to −0.01); p = 0.0356 | −0.08 (−0.2 to 0.03); p = 0.1665 | −0.24 (−0.33 to −0.14); p = 0 |
| **PRISMIII** | 0.5 | 0.74 (0.65 to 0.82) | 0.05 (−0.01 to 0.11); p = 0.1192 | −0.02 (−0.12 to 0.08); p = 0.6285 | −0.02 (−0.12 to 0.09); p = 0.7373 | 0.03 (−0.07 to 0.13); p = 0.603 | 0.05 (−0.06 to 0.15); p = 0.3905 | 0.01 (−0.06 to 0.07); p = 0.8296 | 0.07 (−0.04 to 0.19); p = 0.2264 | **−0.07 (−0.15 to 0);** p = 0.0552 | −0.06 (−0.14 to 0.02); p = 0.1175 | −0.03 (−0.15 to 0.08); p = 0.5461 | −0.03 (−0.15 to 0.09); p = 0.5826 | 0.01 (−0.11 to 0.12); p = 0.9231 | 0.05 (−0.06 to 0.16); p = 0.3698 | −0.11 (−0.17 to −0.04); p = 0.0022 |
| **PRISMIV** | 0.59 | 0.81 | 0.69 (0.59 to 0.79) | −0.07 (−0.17 to 0.03); p = 0.1686 | −0.06 (−0.17 to 0.04); p = 0.2317 | −0.02 (−0.13 to 0.09); p = 0.7161 | 0 (−0.11 to 0.11); p = 0.9897 | −0.04 (−0.13 to 0.05); p = 0.3962 | 0.02 (−0.07 to 0.12); p = 0.6269 | −0.12 (−0.2 to −0.05); p = 0.0016 | **−0.11 (−0.18 to −0.04);** p = 0.0015 | −0.08 (−0.19 to 0.03); p = 0.1445 | −0.08 (−0.19 to 0.03); p = 0.1682 | −0.04 (−0.16 to 0.08); p = 0.4962 | 0 (−0.11 to 0.12); p = 0.9675 | −0.15 (−0.23 to −0.08); p=1e−04 |
| **PIM** | 0.42 | 0.38 | 0.43 | 0.76 (0.67 to 0.85) | 0.01 (−0.03 to 0.04); p = 0.6843 | 0.05 (−0.04 to 0.14); p = 0.2744 | 0.07 (−0.03 to 0.17); p = 0.1564 | 0.03 (−0.09 to 0.15); p = 0.5927 | 0.1 (−0.03 to 0.22); p = 0.1214 | −0.05 (−0.14 to 0.04); p = 0.2868 | −0.04 (−0.12 to 0.05); p = 0.3694 | **−0.01 (−0.06 to 0.04);** p = 0.6812 | −0.01 (−0.07 to 0.05); p = 0.7858 | 0.03 (−0.07 to 0.13); p = 0.5613 | 0.07 (−0.04 to 0.18); p = 0.1885 | −0.08 (−0.16 to 0); p = 0.0486 |
| **PIM2** | 0.35 | 0.31 | 0.38 | 0.94 | 0.75 (0.66 to 0.85) | 0.04 (−0.04 to 0.13); p = 0.3075 | 0.06 (−0.04 to 0.17); p = 0.2195 | 0.03 (−0.1 to 0.15); p = 0.6816 | 0.09 (−0.04 to 0.22); p = 0.1744 | −0.06 (−0.16 to 0.04); p = 0.2726 | −0.05 (−0.14 to 0.05); p = 0.3321 | −0.02 (−0.07 to 0.04); p = 0.5759 | **−0.02 (−0.07 to 0.04);** p = 0.596 | 0.02 (−0.07 to 0.12); p = 0.6349 | 0.07 (−0.05 to 0.18); p = 0.2475 | −0.09 (−0.18 to 0); p = 0.0528 |
| **PIM3** | 0.37 | 0.53 | 0.46 | 0.61 | 0.67 | 0.71 (0.6 to 0.82) | 0.02 (−0.09 to 0.13); p = 0.7227 | −0.02 (−0.13 to 0.09); p = 0.7343 | 0.05 (−0.09 to 0.18); p = 0.5086 | −0.1 (−0.21 to 0.01); p = 0.0837 | −0.09 (−0.2 to 0.02); p = 0.1243 | −0.06 (−0.16 to 0.03); p = 0.2112 | −0.06 (−0.15 to 0.02); p = 0.2039 | **−0.02 (−0.06 to 0.02);** p = 0.3349 | 0.02 (−0.09 to 0.14); p = 0.7018 | −0.13 (−0.23 to −0.03); p = 0.0077 |
| **PELOD** | 0.59 | 0.46 | 0.44 | 0.54 | 0.5 | 0.52 | 0.69 (0.58 to 0.8) | −0.04 (−0.15 to 0.07); p = 0.4866 | 0.03 (−0.09 to 0.14); p = 0.6565 | −0.12 (−0.24 to 0); p = 0.0478 | −0.11 (−0.22 to 0); p = 0.0547 | −0.08 (−0.19 to 0.03); p = 0.1509 | −0.08 (−0.19 to 0.03); p = 0.1636 | −0.04 (−0.16 to 0.08); p = 0.4924 | **0 (−0.09 to 0.1);** p = 0.9485 | −0.15 (−0.26 to −0.05); p = 0.0045 |
| **PELOD2** | 0.43 | 0.77 | 0.59 | 0.28 | 0.23 | 0.48 | 0.48 | 0.73 (0.62 to 0.83) | 0.06 (−0.05 to 0.18); p = 0.2766 | −0.08 (−0.18 to 0.01); p = 0.0972 | −0.07 (−0.18 to 0.04); p = 0.1924 | −0.04 (−0.17 to 0.09); p = 0.5182 | −0.04 (−0.17 to 0.09); p = 0.5433 | 0 (−0.12 to 0.12); p = 0.9768 | 0.04 (−0.06 to 0.14); p = 0.4103 | **−0.11 (−0.2 to −0.03);** p = 0.0082 |
| **PRISM CRP** | 0.81 | 0.45 | 0.63 | 0.4 | 0.31 | 0.37 | 0.56 | 0.5 | 0.66 (0.54 to 0.79) | −0.15 (−0.24 to −0.05); p = 0.0032 | −0.13 (−0.23 to −0.03); p = 0.0084 | −0.11 (−0.22 to 0.01); p = 0.0685 | −0.1 (−0.22 to 0.07); p = 0.0694 | −0.07 (−0.19 to 0.07); p = 0.2944 | −0.02 (−0.11 to 0.07); p = 0.6389 | −0.18 (−0.27 to −0.08); p=3e−04 |
| **PRISMIII CRP** | 0.52 | 0.61 | 0.67 | 0.47 | 0.35 | 0.36 | 0.28 | 0.49 | 0.63 | 0.81 (0.73 to 0.89) | 0.01 (−0.03 to 0.06); p = 0.6393 | 0.04 (−0.05 to 0.13); p = 0.3877 | 0.04 (−0.05 to 0.13); p = 0.3847 | 0.08 (−0.03 to 0.19); p = 0.1557 | 0.12 (0.02 to 0.23); p = 0.0202 | −0.03 (−0.07 to 0.01); p = 0.1187 |
| **PRISMIV CRP** | 0.65 | 0.57 | 0.73 | 0.54 | 0.45 | 0.34 | 0.36 | 0.37 | 0.6 | 0.85 | 0.8 (0.72 to 0.88) | 0.03 (−0.05 to 0.11); p = 0.4858 | 0.03 (−0.06 to 0.12); p = 0.502 | 0.07 (−0.04 to 0.18); p = 0.2262 | 0.11 (0 to 0.22); p = 0.0476 | −0.04 (−0.1 to 0.01); p = 0.1203 |
| **PIM CRP** | 0.53 | 0.27 | 0.37 | 0.88 | 0.81 | 0.59 | 0.44 | 0.19 | 0.49 | 0.52 | 0.61 | 0.77 (0.67 to 0.87) | 0 (−0.04 to 0.04); p = 0.9526 | 0.04 (−0.05 to 0.13); p = 0.3739 | 0.08 (−0.03 to 0.2); p = 0.1457 | −0.07 (−0.15 to 0.01); p = 0.0794 |
| **PIM2 CRP** | 0.49 | 0.19 | 0.33 | 0.79 | 0.82 | 0.62 | 0.43 | 0.15 | 0.5 | 0.5 | 0.54 | 0.9 | 0.77 (0.67 to 0.87) | 0.04 (−0.04 to 0.12); p = 0.3386 | 0.08 (−0.03 to 0.19); p = 0.1468 | −0.07 (−0.16 to 0.01); p = 0.0975 |
| **PIM3 CRP** | 0.47 | 0.43 | 0.39 | 0.54 | 0.58 | 0.93 | 0.48 | 0.39 | 0.48 | 0.43 | 0.41 | 0.67 | 0.74 | 0.73 (0.61 to 0.85) | 0.04 (−0.07 to 0.16); p = 0.449 | −0.11 (−0.21 to −0.02); p = 0.0216 |
| **PELOD CRP** | 0.45 | 0.42 | 0.4 | 0.4 | 0.36 | 0.44 | 0.63 | 0.56 | 0.69 | 0.44 | 0.35 | 0.4 | 0.42 | 0.49 | 0.69 (0.58 to 0.79) | −0.15 (−0.24 to −0.07); p=4e−04 |
| **PELOD2 CRP** | 0.53 | 0.66 | 0.64 | 0.53 | 0.42 | 0.51 | 0.4 | 0.59 | 0.64 | 0.88 | 0.76 | 0.59 | 0.52 | 0.58 | 0.62 | 0.84 (0.77 to 0.91) |

**Notes.**

[a]Completed data: 246 patients; AUC with CI for each score on diagonal, correlation between scores below diagonal, difference in AUC with CI and *p* value above diagonal.

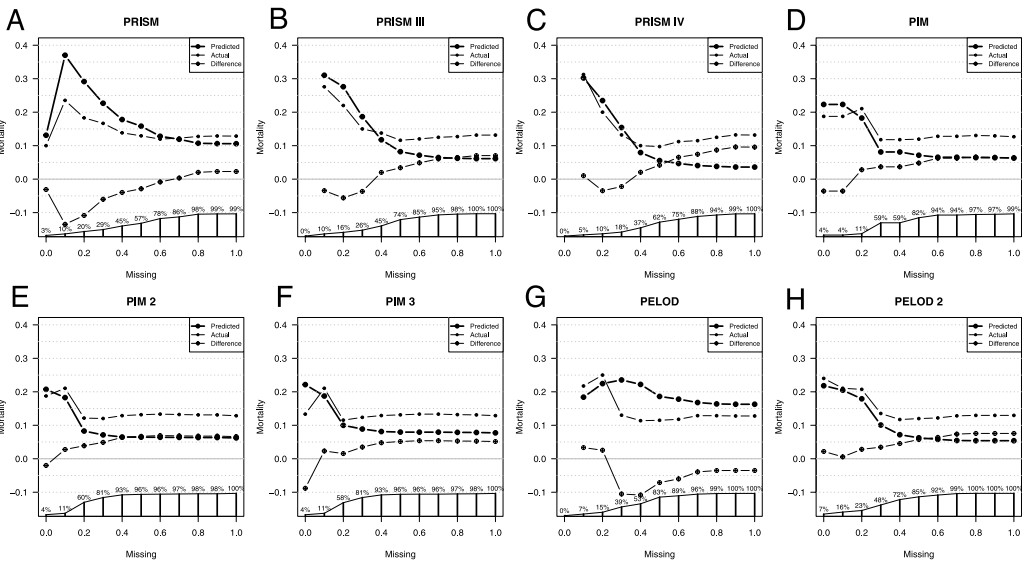

**Figure 2  Influence of missing data on score performance.** Predicted (large dots) and actual (small dots) mortality including the difference (x in a circle) depending on the extent of missing values for scoring of the Pediatric Risk of Mortality scores PRISM (A), PRISM III (B), and PRISM IV (C), the Pediatric Index of Mortality scores PIM (D), PIM 2 (E), and PIM 3 (F), as well as the Pediatric Logistic Organ Dysfunction scores PELOD (G), and PELOD 2 (H). The proportion of patients available for analysis from the total population is given in percentage along the x axis in A–H.

The AUC of the scores changes only minimally as a consequence of the number of missing values. Excluded here is PIM3, whose AUC with the completeness of the parameters is lower than the AUCs with missing values.

The PRISM score had a difference of 0.1 in the AUCs with the highest AUC of 0.66 at 50% missing values allowed. PRISMIII and PRISMIV had a difference of 0.11 in the AUCs. PRISMIII and PRISMIV had their highest AUCs of 0.84 at 30% missing values allowed. Also, the PELOD score had a difference of 0.11 in its AUCs, calculated according to the degree of missing values, with the highest AUC of 0.74 at 30%–40% missing values allowed.

PIM and PIM2 had a difference of only 0.05 in their AUCs and had their highest AUC at 30%–40% and 20% missing values allowed. Also, PELOD 2 had a small difference of 0.07 with its highest AUC at 30% missing values allowed.

# DISCUSSION

The aim of this study was to investigate and compare various common mortality risk assessment scoring systems, namely PRISM, PRISM III, PIM, PIM2, PIM3, PELOD and PELOD2 in pediatric sepsis patients. In doing so, we also evaluated different time points for the score assessments, namely PICU admission and the day of C-reactive protein peak. Furthermore, we investigated the influence of missing parameters on the predictive power of the scores.

## Comparison of the scores at admission

The mortality rate in our study was at 13.6% in the midfield of other PICUs in developed countries (*Ruth et al., 2014*). The difference between the predicted and the actual mortality of the individual scores in our septic patient population is roughly comparable to that of other studies (*Arias Lopez et al., 2018*; *Dewi et al., 2019*; *Hamshary et al., 2017*; *Patki, Raina & Antin, 2017*; *Qiu et al., 2017a*; *Qureshi, Ali & Ahmad, 2007*; *Taori, Lahiri & Tullu, 2010*).

The PIM scores (PIM1, PIM2, PIM3) underestimate overall mortality as compared to actual mortality, as confirmed by other studies (*Patki, Raina & Antin, 2017*; *Qiu et al., 2017a*; *Taori, Lahiri & Tullu, 2010*). By contrast, the PRISM score, with its mortality prediction, gives quite a good comparison of the actual mortality of the entire population. This has also been confirmed in other studies (*Qiu et al., 2017a*; *Taori, Lahiri & Tullu, 2010*). While PELOD showed a slight overprediction in mortality in our population, PELOD 2 showed a significant underprediction of the observed mortality. An even greater discrepancy was found in a study by *Goncalves et al. (2015)*.

Nevertheless, when looking at the ability to predict mortality for the individual patient, the PIM 2 score shows the best performance as reflected by highest AUC, closely followed by PIM and PIM3 but also by PRISM III and PELOD 2. Other studies also found a slightly higher AUC in PIM2 than in PIM (*Brady et al., 2006*; *Slater & Shann, 2004*). Even the good performance of PRISM III and PELOD 2 for the individual mortality prediction was also shown by Gonçalves et al. in a general critically ill pediatric population (*Gonçalves et al., 2015*).

We found that the PRISM score to be the worst performer (AUC 0.63) in our septic population. In contrast to our findings, a prospective study conducted in pediatric patient populations of specialist multidisciplinary ICUs showed the AUC (0.90) of the PRISM score to be clearly higher than our result (*Slater & Shann, 2004*). In another study of children with meningococcal sepsis, the PRISM score even outperformed the PIM score (*Leteurtre et al., 2001*).

There are various ongoing discussions as to whether newer versions of the individual scores will improve the predictive value. While a multicenter study in Italy confirmed a significant improvement in PIM 3 compared to PIM 2 (*Wolfler et al., 2016*), *Tyagi, Tullu & Agrawal (2018)* found no relevant improvement between PIM 2 and PIM 3. We were able to determine an increase in the predictive value of the scores along versions, but the last versions (PIM 3 and PRISM IV) brought no further improvement.

## Comparison of the scores recorded at the time of the C-reactive protein peaks

The scores are intended for a broad mass, meaning all kinds of diseases and conditions. For each score, a specific point in time or timeframe was determined, for which the best performance is to be expected. While the PRISM and PRISM III scores are calculated after 24 h in-hospital, the PIM scores are computed within the first hour after admission. One drawback of a 24-hour assessment is that the patient may already be dead before the score can give a prognosis. In the case of an assessment made in the first hour, however, there is an inaccuracy factor regarding preclinical care. In seriously ill, well-cared-for and stabilized

patients, a score may be deceptively low in value. As of version PIM 2, an attempt was made to compensate this with an additional parameter ("main reason for ICU admission").

In septic patients there is a similar problem: in some cases sepsis was the reason for admission, while in other cases sepsis developed during the course of hospital stay, possibly in postoperative patients (*Sidhu et al., 2015*; *Wang et al., 2018*). In such a case, sepsis cannot be predicted at the time of admission and thus at the time of data collection. A score calculated during the most severe septic phase would therefore show better performance. Thus, our clientele's scores were reassessed at peak C-reactive protein. This analysis revealed that with disease progression PRISM IV and PELOD 2 were becoming significantly more precise in predicting mortality. We conclude that for PRISMIV and PELOD 2 the time when evaluation is performed is important for mortality prediction, while for the other scores the time of evaluation has no significant influence on the predictive ability of these scores. Just as *Leteurtre et al. (2015)* we feel that especially the PELOD 2 score serves well to monitor the progression of disease severity and predict outcome when evaluated regularly during the course of the disease.

## Comparison of scores for missing parameters

Although it is difficult, even in a prospective setting, to collect all the necessary data for creating the scores (*Tibby et al., 2002*), it is even more difficult with a retrospective study design. Parameters are not or not fully recorded, not available at the specified time, not collected or lost due to incomplete documentation. However, this reflects the realism of everyday life.

This problem has already been addressed by the developers of the PRISM score, who concluded that the missing values are often normal and therefore will hardly influence the score (*Pollack et al., 1996a*). The same assumption that missing values are normal was partially implemented in the scoring validation studies (*Gorges et al., 2018*; *Leclerc et al., 2017*). It was also incorporated into the PIM score, so that it is possible to specify missing data as such and thus there is no change in score points, which makes sense to a certain extent. For example, a lactate level that describes tissue hypoxia may not have been subject to lab testing by the treating physician, because the patient's medical condition was not presumed to be so poor. The situation is similar for other parameters.

Nevertheless, this assumption might be misleading: for example, there is only a small blood volume available for laboratory testing, especially in young pediatric patients (*Sztefko et al., 2013*). This might be supported by a validation study, where PELOD 2 and PRISM III scores show decreased performance when it is assumed that the unavailable data are within normal ranges (*Gorges et al., 2018*).

We were able to show that only in a few cases was it possible to retrospectively collect 100% of the data for scoring. Here only a small patient population remained, so that the analyses could no longer be performed validly. However, it was seen that patients with a low percentage of missing values have high mortality.

With increasing data availability predicted and actual mortality approached each other, similar to what Agor and his team found in their study of the impact of missing scores on adult scores (*Agor et al., 2019*). For our patients, the predicted and the actual mortality

were quite close, except for the PRISM, PELOD and PIM3 scores, where the difference between the predicted and the actual mortality fluctuated, especially when only few missing values were allowed.

The most stable scores in terms of missing values, defined as the maximum deviation from predicted and actual mortality, have been shown to be PRISM III, PIM and PIM 2. Here, it has to be mentioned that, when a high percentage of missing values is allowed, mortality is underestimated by the scores, while with increasing data availability the scores tend to overestimate. The exceptions here are the PELOD score where the results are converse and PELOD 2 score, which consistently slightly overestimates mortality.

The AUC of the scores, however, changes only minimally with the number of missing values allowed. The smallest difference in the AUCs depending on the allowed missing values was seen in the PIM and PIM2 scores as well as the PELOD 2 score, which thus proved to be stable as compared to the missing values.

### Limitations

The retrospective study design is limited in terms of score performance because some patients did not have all the data needed to calculate the scores. Since this is a single-center study, the number of patients needed for a valid statistical analysis is low overall, especially in the group of patients with 100% availability of the required data. In our study we assessed only the average effect of the missing values and not the weighting of the individual missing parameters necessary for the score. On the other hand, it was possible for us to draw a realistic scenario of data availability in connection with the score survey in a retrospective study design.

## CONCLUSION

The results demonstrate that at ICU admission all PIM scores together with PRSIM III and PELOD 2 are the most suitable scores for mortality prognosis, while the PRISM score is the worst. Once the sepsis is pronounced, PELOD 2 and PRISM IV became significantly better in their performance and count among the best prognostic scores for use at this time. Therefore, when calculated at multiple times, the PELOD 2 score is best suited for assessing the prognosis on an ongoing basis.

Overall, PIM scores show comparatively good performance, are stable as far as timing of the disease survey is concerned, and they are also relatively stable in terms of missing parameters. PELOD 2 is the best for monitoring and is also relatively stable in relation to missing values.

### Abbreviations

| | |
|---|---|
| **aPTT** | activated thromboplastin time |
| **AUC** | area under the curve |
| **CI** | confidence interval |
| **ICU** | Intensive Care Unit |
| **PELOD** | Pediatric Logistic Organ Dysfunction |
| **PICU** | Pediatric Intensive Care Unit |

| | |
|---|---|
| **PIM** | Pediatric Index of Mortality |
| **PRISM** | Pediatric Risk of Mortality |
| **PT** | prothrombin time (Quick) |
| **ROC** | receiver operating characteristic |

## ACKNOWLEDGEMENTS

We thank Dr. Katharina Auer and Dr. Sophie Hofer, who helped with data acquisition, and Dr. Dietmar Fries for his assistance in conducting this project.

### Funding

The authors received no funding for this work.

### Competing Interests

The authors declare that the research was conducted in the absence of any commercial or financial relationships that could be construed as a potential conflict of interest.

Mirjam Bachler has received personal fees and travel grants from LFB Biomedicaments, Takeda GmbH, CSL Behring GmbH, Mitsubishi Tanabe and non-financial support from TEM International outside the submitted work.

### Author Contributions

- Christian Niederwanger conceived and designed the experiments, analyzed the data, authored or reviewed drafts of the paper, and approved the final draft.
- Thomas Varga conceived and designed the experiments, authored or reviewed drafts of the paper, and approved the final draft.
- Tobias Hell analyzed the data, prepared figures and/or tables, authored or reviewed drafts of the paper, and approved the final draft.
- Daniel Stuerzel, Jennifer Prem, Magdalena Gassner, Franziska Rickmann, Christina Schoner, Daniela Hainz, Gerard Cortina, Benjamin Hetzer and Benedikt Treml performed the experiments, authored or reviewed drafts of the paper, and approved the final draft.
- Mirjam Bachler conceived and designed the experiments, performed the experiments, analyzed the data, prepared figures and/or tables, authored or reviewed drafts of the paper, and approved the final draft.

### Ethics

The following information was supplied relating to ethical approvals (i.e., approving body and any reference numbers):

The study protocol was approved by the Institutional Review Board of the Medical University of Innsbruck (AN2013-0044 and EK Nr: 1109/2019).

### Data Availability

The raw measurements are available in the Supplemental Files.

## Supplemental Information

Supplemental information for this article can be found online at http://dx.doi.org/10.7717/peerj.9993#supplemental-information.

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
