# Peer review of "Comparison of pediatric scoring systems for mortality in septic patients and the impact of missing information on their predictive power: a retrospective analysis"

_PeerJ, doi:10.7717/peerj.9993_

## Round 0.1 · original submission · Major Revisions

Dear authors,

After reading the reviewers' comments, I think your manuscript has scientific validity to be published in PeerJ, once some issues highlighted in their reports are solved by you. Please, see their comments below this email.

Best regards,
Dr Palazón-Bru (academic editor for PeerJ)

Reviewer 1 ·

Basic reporting

In their study “Comparison of pediatric scoring systems for mortality in septic patients and the impact of missing information on their predictive power” Christian Niederwanger and his colleagues evaluated different scoring systems used to assess and monitor children with sepsis. In their retrospective study the authors evaluated the medical records of 398 patients under 18 years of age with sepsis and apllied the following scoring systems: PRISM, PRISM III, PRISM IV, PIM, PIM2, PIM3, PELOD, PELOD 2. They found that in particular the PIM scores are helpful in assessing the severity and that missing data do not seem to compromise the prognostic power.
The article is structured according to the guideline of the journal. The literature cited is extensive. The abstract summarizes the important results of the study. The introduction is brief and informative and the statistical methods seem appropriate. Limitations are addressed. Figures and tables are informative.

Experimental design

The research questions are well defined. The experimental design (retrospective review of existing medical data of a single institution and application established scoring systems) is straight forward but as eluded to in the limitiation flawed by the nature of a retrospective design and in particular incomplete data sets. It is interesting that the authors tackle this problem from the start head on and included the problem of incomplete data sets in their statistical analyses.

As far as can tell most scoring systems have been evaluated in much larger studies and in a prospective fashion so the overall novelty of the findings is limited.

Validity of the findings

The authors provided all the necessary data and discussed their findings in the context of the existing literature. Open questions are addressed.
The conclusion is very lengthy and should be shortened

Additional comments

I have several further remarks:

• The article is overall well written and to the point apart from several wording issue and should be read by a native speaker.
(see Introduction for example, line 81: However, some patients develop certain life-threatening clinical pictures- please change to conditions etc;
line 100 Please change as follows: Also, the optimal timing for determining these scores in the clinical course picture of sepsis was evaluated in addition to the influence of the lack of data on the predictive value of the different scores.

• Methods: the definition of children with sepsis should be briefly described.

• The discussion is lengthy and could be more concise

• Line 248: the authors compare the finding of their study with a large prospective australien prospective cohort study (Slater and Shann, 2004) which is not entirely correct and should be rephrased.

Reviewer 2 ·

Basic reporting

no comment

Experimental design

no comment

Validity of the findings

no comment

Additional comments

In this study, the authors evaluated and compared the prognostic ability of several common pediatric scoring systems to determine which is the most applicable score for pediatric sepsis patients from timing of disease survey and insensitivity to missing data. The main findings were:
They found that PIM scores show comparatively good performance, are stable as far as timing of the disease survey is concerned, and the system is also relatively stable in terms of missing parameters. And PELOD 2 is best suitable for monitoring clinical course.
The work in general has been well carried out, and the results give a clear hypothesis regarding the pediatric scoring systems.

·

Basic reporting

no comment

Experimental design

Please explain the relationship between peak CRP and disease severity.
Please provide a flow chart in the methods section.

Validity of the findings

Whether the weight of missing data affects the predicted results more than the number of missingdata?

Additional comments

Overall the quality of the article is fairly good and provides the effect of missing data on the prediction results, which has clinical practical significance.

---

## Round 0.2 · accepted · Accept

All the reviewers' concerns have been correctly addressed.